# *Cryptomphalus aspersa* Egg Extract Protects against Human Stem Cell Stress-Induced Premature Senescence

**DOI:** 10.3390/ijms25073715

**Published:** 2024-03-27

**Authors:** Zozo Outskouni, Christina Christodoulou, Andreas Goutas, Ioannis D. Kyriazis, Adamantini Paraskevopoulou, George P. Laliotis, Anthia Matsakidou, Athanasios Gogas, Varvara Trachana

**Affiliations:** 1Department of Biology, Faculty of Medicine, University of Thessaly, 41500 Larisa, Greece; zooutskouni@uth.gr (Z.O.); christichris96@gmail.com (C.C.); agoutas@bioacademy.gr (A.G.); ioankyriazis@uth.gr (I.D.K.); 2Biomedical Research Foundation of the Academy of Athens, 11527 Athens, Greece; 3Laboratory of Food Chemistry & Technology, School of Chemistry, Aristotle University of Thessaloniki, 54124 Thessaloniki, Greece; adparask@chem.auth.gr (A.P.); matsakidou@chem.auth.gr (A.M.); 4Laboratory of Animal Breeding and Husbandry, Department of Animal Science, Agricultural University of Athens, 75 Iera Odos, 11855 Athens, Greece; glaliotis@aua.gr; 5Le Blanc Skincare, 41222 Larisa, Greece; atgogas@gmail.com

**Keywords:** mesenchymal stem cells, senescence, snail egg extract, nutraceuticals

## Abstract

Cellular senescence is a tightly regulated pathophysiologic process and is caused by replicative exhaustion or external stressors. Since naturally derived bioactive compounds with anti-ageing properties have recently captured scientific interest, we analysed the anti-ageing and antioxidant efficacy of *Cryptomphalus aspersa* egg extract (CAEE). Its effects on stemness, wound-healing properties, antioxidant defense mechanisms, and DNA damage repair ability of Human Wharton’s jelly mesenchymal stem cells (WJ-MSCs) were analysed. Our results revealed that CAEE fortifies WJ-MSCs stemness, which possibly ameliorates their wound-healing ability. Additionally, we show that CAEE possesses a strong antioxidant capacity as demonstrated by the elevation of the levels of the basic antioxidant molecule, GSH, and the induction of the NRF2, a major antioxidant regulator. In addition, CAEE alleviated cells’ oxidative stress and therefore prevented stress-induced premature senescence (SIPS). Furthermore, we demonstrated that the prevention of SIPS could be mediated via the extract’s ability to induce autophagy, as indicated by the elevation of the protein levels of all basic autophagic molecules and the increase in formation of autophagolysosomes in CAEE-treated WJ-MSCs. Moreover, CAEE-treated cells exhibited decreased Caveolin-1 levels. We propose that *Cryptomphalus aspersa* egg extract comprises bioactive compounds that can demonstrate strong antioxidant/anti-ageing effects by regulating the Caveolin-1–autophagy–senescence molecular axis.

## 1. Introduction

Mesenchymal stem cells (MSCs) are multipotent cells with the ability to differentiate into mesodermal lineage cell types and a capacity for self-regeneration [1]. MSCs populations derived from bone marrow were first identified in 1974 [2]. Since then, MSCs have been isolated from various adult tissues, such as adipose tissue, muscles, and dental pulp [3,4,5]. Nowadays, MSCs have also been isolated from fetal or perinatal tissues, including the umbilical cord and the amniotic fluid [6]. Although these populations share some characteristics, MSCs derived from perinatal tissues exhibit some unique and valuable properties. 

Wharton’s jelly (WJ-MSCs) cells [7] are MSCs derived from the connective tissue of the umbilical cord that exhibit some of the properties of embryonic stem cells (ESCs), such as the high-proliferative capacity without potential safety concerns, i.e., the risk of teratoma formation, while possessing a broad differentiating capacity and lower immunogenicity, compared to their adult counterparts [6]. These characteristics, accompanied by minor ethical concerns and the low cost of their isolation procedures, have deemed them an invaluable tool for regenerative medicine and stem cell therapy [8]. 

Nonetheless, cell-based therapies require a significant number of cells and therefore their prolonged in vitro expansion. This springs out as a significant limitation of the clinical use of MSCs, as prolonged culture would inevitably lead to cellular senescence [9]. 

During cellular senescence, cells undergo an irreversible cell-cycle arrest and exhibit several distinct features, such as an enlarged morphology, accumulation of DNA damage, and elevated levels of cell-cycle inhibitors. However, they remain metabolically active, secreting a plethora of factors which constitute the senescence-associated secretory phenotype (SASP) [1]. Stem cell senescence influences their proliferative capacity and differentiation potential, hindering their regenerative and homeostatic abilities, and therefore complicating their clinical use [9]. So, the development of approaches that could delay cellular senescence would prove beneficial for cell-based therapies.

The imbalance between reactive oxygen species (ROS) formation and the antioxidant defense system, known as oxidative stress (OS) [10], is a major regulator of cellular senescence. OS can lead to stress-induced premature senescence (SIPS) [11] through the accumulation of genomic damage and disruption of proteostasis [12]. Hence, intensive research of antioxidants and new anti-ageing agents is of paramount importance to prevent or alleviate cellular dysfunction resulting from senescence [13]. 

Even more, over the last decades, numerous preclinical investigations and various interventions have aimed at delaying the onset of ageing and lengthening healthy lifespan. Several pathways have been identified as crucial factors in regulating ageing and lifespan, turning the implicated molecules into possible anti-ageing targets [14,15]. In addition to lifestyle changes and conventional pharmacological interventions, nutraceuticals (naturally derived compounds with a wide range of biological effects) have been extensively used to delay the signs of ageing and alleviate a variety of age-related manifestations [16]. These bioactive components could be derived from plants, herbs, or natural foods and their byproducts [17]. *Cryptomphalus aspersa* (syn. *Helix aspersa*), the common garden snail, has been identified as a source of such compounds that recently received extensive attention from the healthcare and cosmetic industry.

Snails belong to the phylum Mollusca and class Gastropoda. They are predominantly aquatic, but some are terrestrial. Among terrestrial gastropods, Helix snails are a famous and well-described genus [18]. Members of this genus are found in Europe, Mediterranean North Africa, and western Asia [19] and comprise four popular edible snail species: *Helix aspersa* (*Cryptomphalus aspersa*) and *Helix pomatia* (Escargot de Bourgogne), *Helix lucorum* (Turkish snail), and *Helix vermiculata* (*Eobania vermiculata*) [20]. Snail meat and eggs are considered delicacies and are characterised by excellent nutrient traits because of their high protein, low calories, and rich mineral and low fat content [21,22]. 

In addition to snail farming, mucus extract has received considerable attention as an attractive candidate for medical products concerning wound management [23] and as a component of cosmetic products with anti-ageing effects. Snails secrete mucus for vital purposes. The mucus from *Cryptomphalus aspersa* is a lubricant with moisturizing, antimicrobial [24,25], anti-inflammatory [26], and healing capabilities [27,28]. These characteristics are derived from mucus bioactive compounds, such as allantoin glycolic acid, collagen, vitamins, minerals, and several antioxidant molecules [18,29]. Brieva and colleagues shed light on the molecular basis of the regenerative properties of *Cryptomphalus aspersa* secretion, displaying its antioxidant superoxide dismutase (SOD) and glutathione S-transferase (GST) activities, as well as its stimulatory effect on extracellular matrix assembling and actin cytoskeleton rearrangement [30]. Other in vitro studies indicate its ability to promote cell survival, trigger dermal fibroblast and keratinocyte proliferation and migration [31], prevent oxidative damage [32], and diminish fibroblast ageing [33]. 

Although mucus has been the center of attention for both researchers and pharmaceutical industries, little is known about the medical properties of other snail products, such as the eggs. Espada and colleagues demonstrated that *Cryptomphalus aspersa* egg extract contributes to skin restoration due to its improving effect on a series of functions related to cell migration and mitigated cutaneous cell-age-related morphology [34]. It also provides protection from telomere shortening, a feature responsible for replicative senescence [35]. Additionally, egg extract contributed to mesenchymal stem cells differentiation to keratinocytes and fibroblasts and exhibited a preventive or repairing role when tissue is damaged either by external or internal causes [36]. In contrast to mucus extract, egg extract seems to decelerate proliferation in favour of the mobilization of genome integrity surveillance mechanisms [35]. Considering its anticancer qualities [37] Matusiewicz and colleagues point out the negative effect of egg extract on Caco-2 colon cancer cells’ viability, paving the way for a combinatorial with conventional methods, and thus more compelling, cancer therapy [36].

Given all the above, in this study we aimed at investigating the possible beneficial properties of *C. aspersa* egg extract on WJ-MSCs. To that end, we assessed whether the egg extract has the ability to enhance cell migration as well as WJ-MSCs’ antioxidant capacity. Furthermore, we aimed to assess its effect on oxidative DNA damage repair. Most importantly, we investigated whether the egg extract has a protective ability against stress-induced premature senescence (SIPS). Finally, we also assessed how *C. aspersa* egg extract affects caveolin-1 as well as autophagy in order to provide a mechanistic insight considering its bioactive abilities against OS. 

## 2. Results

### 2.1. Cryptomphalus Aspersa Egg Extract (CAEE) Accelerates Wound Healing

First, we opted to assess CAEE’s potential beneficial effects on would healing since this is an effect reported for other parts of the snail [26,28,31]. To that end, we performed a wound-healing assay (or cell-migration assay). The method mimics cell migration during wound healing in vivo, as a linear thin scratch (creating a gap), a “wound”, is created in a confluent cell monolayer [38]. MSCs “wound-” healing ability was tested in the presence of various concentrations of CAEE or in their regular culturing medium. Images were taken at two time points; after “wound” infliction (0 h) and after 24 h, as depicted in (Figure 1 and Appendix A). At 24 h, WJ-MSCs left to recover in CAEE had a significantly higher “wound-” closure percentage at all different concentrations of the extract compared to their untreated counterparts, suggesting that the extract promotes cell migration and possibly possesses wound-healing ability. In particular, CAEE treatment at 0.1 mg/mL resulted in the highest wound area closure percentage (71.2 ± 2.52%) as compared to the untreated control (30.71 ± 2.52%), and this concentration was henceforth used in the rest of the experiments. It should be mentioned that similar [34] or even lower [39] concentrations were previously used in different experimental set-ups and demonstrated beneficial effects. 

### 2.2. CAEE Increases the Expression of Stemness Regulator OCT4 

Driven by the insight that *C. aspersa* egg extract possesses the ability to induce differentiation in adipose-derived MSCs [40] and that stem cells derived from the umbilical cord express stemness and pluripotency regulator factors much like embryonic stem cells [41], we investigated whether CAEE exerted any effect on these stemness regulator factors. We have found that treatment with 0.1 mg/mL of CAEE led to increased OCT4 protein levels (Figure 2).

This insight is important for the understanding of the biological effect of CAEE and its constituents since OCT4 plays a central role in stemness and the stimulation of cellular pathways that are critical for wound healing and tissue rejuvenation [42]. 

### 2.3. CAEE Augments the Antioxidant Potential of WJ-MSCs

Subsequently, we assessed possible antioxidant effects of CAEE by determining total glutathione (GSH) levels, as GSH is a basic factor representing the antioxidant cellular potential [43]. Our results showed that cells treated with CAEE for 24 h resulted in a 30% elevation in GSH levels compared to not-treated WJ-MSCs (Figure 3A). Moreover, the effect of CAEE on lipid peroxidation was also studied through the thiobarbituric acid reactive species assay (TBARS). Our results showed a 28.7% decrease in TBARS levels after 24 h of treatment with CAEE (Figure 3B), further strengthening our observations regarding the extract’s potent antioxidant capacity. 

Subsequently, apart from the GSH and TBARS levels, we examined the effect of CAEE on the protein levels of the major antioxidant regulator nuclear factor erythroid 2 related factor 2 (NRF2) and its basic transcriptional target NADPH quinone oxidoreductase 1 (NQO1), a significant antioxidant enzyme [44,45]. Our results indicated a significant increase in both NRF2 and NQO1 protein levels after 24 h of CAEE treatment (Figure 3C,D), suggesting that CAEE possesses strong antioxidant properties that could be possible mediated through the NRF2-related pathways.

### 2.4. CAEE Promotes DNA Damage Repair

To further explore the augmented antioxidant potential of CAEE, we assessed whether the treatment with the extract could have a beneficial impact against DNA damage formation. During cell culture, MSCs accumulate DNA damage due to both extrinsic and intrinsic stressors but also due to the dysregulation of the antioxidant machinery and consequent elevation in ROS levels [12]. Since in our experimental approach cells were in a middle passage, we speculated that they would have accumulated DNA damage to a certain extent. To examine that, cells were treated or not for 24 h with CAEE, and DNA damage formation (double-strand breaks; DSB) was assessed by examining the presence of 53BP1 foci, a protein involved in DSB repair [46], microscopically. Our results showed that incubation with CAEE significantly reduced the percentage of cells positive for 53BP1 foci (15.29 ± 0.64%) as compared to the untreated cells (26.20 ± 2.46%) (Figure 4). This implies that CAEE treatment can promote the repair of oxidative stress-induced DNA damage.

### 2.5. CAEE Protects against Stress-Induced Premature Senescence (SIPS) 

It remains crucial to assess whether CAEE can protect against premature cellular senescence that is induced by excessive oxidative stress [47,48]. For that, WJ-MSCs were treated with CAEE and subsequently exposed to H_2_O_2_ (400 μM for 2 h) to induce premature senescence (SIPS). Senescence was assessed by evaluating the activity β-galactosidase enzyme (SA-β-gal; Figure 5A,B). As indicated, cells treated with CAEE prior to SIPS induction appear to have significantly less SA-β-gal positive cells (blue), which implies that they were protected from premature senescence. Analysis revealed that the percentage of SA-β-gal positive cells treated with the extract prior to senescence induction was significantly lower (CAEE SIPS p2 = 12.88 ± 1.05%) than that in untreated cells representing the control experimental group (cells not treated with the extract prior to SIPS induction; SIPS p2 = 76.93 ± 4.5%). 

In order to further validate the effect of CAEE on the onset of premature senescence due to oxidative stress, we also evaluated p21 and p16 protein levels, which are cell-cycle inhibitors and senescence markers [49,50]. As expected, the protein levels for both p21 and p16 were significantly reduced in cells treated with the extract prior to SIPS induction compared to untreated cells (Figure 5C,D). 

Taken together, we can conclude that CAEE protects WJ-MSCs from premature senescence that is triggered by oxidative stress.

### 2.6. CAEE Decreased Caveolin-1 Protein Levels

Since CAEE treatment was able to prevent SIPS, we next opted to assess whether CAEE has an impact on Caveolin-1 expression, a protein that participates in the regulation of senescence onset in WJ-MSCs, as we have previously reported [51,52]. As expected, our results showed that Caveolin-1 protein levels were decreased in WJ-MSCs treated with CAEE compared to the not-treated control cells (Figure 6).

### 2.7. CAEE Protective Effect against SIPS Is Associated with Autophagy Induction

Autophagy is a process with a strong connection to oxidative stress and an ambiguous relationship with cellular senescence. Furthermore, Caveolin-1 is implicated in the autophagic pathway as the recent literature has revealed [51,53,54,55]. For that reason, the protein levels of key molecules involved in various stages of the autophagic pathway were assessed in cells treated with CAEE prior to SIPS induction and were found to be increased compared to cells that were not treated with CAEE prior to senescence induction (SIPS p2; Figure 7). Specifically, ULK-1, PI3K-CIII, and Beclin-1 that are involved in the upstream stages of autophagy induction and autophagosome formation were increased. The levels of ATG5 were also increased followed by an increase in LC3-II/I ratio and a concurrent decrease in p62, which is a typical autophagy substrate, suggesting that autophagic flux is induced in the presence of CAEE.

As autophagy induction could also be assessed microscopically, we analysed the LC3 staining pattern with immunofluorescence. As we [56] and others [57,58] have previously demonstrated, LC3 localises to the autophagosome membrane, and its appearance as punctate structures characterises autophagy stimulation. Indeed, treatment with CAEE results in autophagosome formation as indicated by the characteristic dotted staining of LC3 (Figure 8A) and the increase in the percentage of cells positive for LC3 puncta (89.12%) compared to the untreated cells (34.10%) (Figure 8B). Furthermore, the induction of autophagy due to CAEE treatment is also indicated by the diffuse p62 staining, suggesting its degradation by autophagy (Figure 8A). 

Taken together, CAEE affected the function of autophagy, revealing a new possible molecular axis in which CAEE is able to exert its protective role against SIPS.

## 3. Discussion

Ageing is an intricate degenerative biological process culminating in the impairment of functionality in several tissues in a time-dependent manner [59]. Cellular senescence, a state in which cells stably exit the cell cycle, leads to the loss of proliferative capacity and is closely related to organismal ageing [60]. Senescence can be the result of both extrinsic and intrinsic factors and impacts stem cells as well. Although it is speculated that cellular senescence evolved as a mechanism aiding in the protection against tumorigenesis, it is the driving force behind many pathologies and age-related diseases [61]. Therefore, many strategies are currently being developed to attenuate it. One such approach is the search for bioactive compounds derived from natural sources called nutraceuticals [59]. The advantage of such compounds or extracts is that they have unmatched chemical diversity with enhanced structural complexity. It is moderately estimated that almost 40% of the nature-derived compound diversity cannot be met with our available technological tools and generated in our synthetic compounds libraries. Moreover, compounds derived from natural sources are molecules that have co-evolved in complex biological environments offering specific biological interaction. Therefore, there is an augmented possibility that most of the chemical compounds that are present in a naturally derived extract can easily interact with proteins and modulate their function.

Mesenchymal stem cells (MSCs) have emerged as pivotal players in the realm of biomedicine since they possess unique properties. Their versatility in terms of origin and the different cellular types that they can generate via their differentiation make them invaluable in regenerative medicine. Apart from this, MSCs are used to treat autoimmune disorders, graft-versus-host disease, or even COVID-19-related inflammatory complications, owing to their immunomodulatory capabilities Additionally, several researchers have described a promising tumor-suppression ability of MSCs, and especially those derived from fetal or perinatal tissues, such as Wharton jelly MSCs [62]. All the above render MSCs a promising platform to study cell behaviour and interactions in disease-based controlled environments. However, as mentioned, their prolonged in vitro culture, which is necessary in order to reach a sufficient cell population, results in senescence that distorts their actual ability for cell therapies. For that, researchers need to enrich the scientific literature with insights that enhance or even protect them from the technical limitations that arise during their laboratory use. 

Our study aimed to offer middle-passage MSCs maintenance of their ability to heal wounds, preserve balanced redox status, and delay or avoid cellular senescence. For that, we investigated the effect of *C. aspersa* egg extract (CAEE). 

Proliferative and pro-migratory effects of snail mucus have been previously reported [28,30,63], but investigation of such qualities regarding snail eggs remains limited, with only a small number of available studies involving different types of cells [34,39]. One important report indicated that CAEE is able to accelerate the wound-healing process in human dermal papilla stem cells (HHDPCs) [39]. WJ-MSCs’ ability to heal wounds has been attributed to their paracrine secretion of wound-healing promoting factors, such as VEGF and TGF-β, among others [64,65,66]. The wound-healing potential of CAEE, as demonstrated in the cell-migration assay, reported here could be mediated via paracrine actions of WJ-MSCs resulting in the regulation of molecules and mechanisms involved in cell migration and, ultimately, tissue regeneration. 

Stemness, a unique property of stem cells, has a cornerstone role in the intricate process of wound healing. Several molecular factors orchestrate the regenerative response, such as OCT4, SOX2, and NANOG, that are able to direct the pluripotency of stem cells and contribute to tissue repair [42]. Wnt-, notch-, and hedgehog-related pathways regulate fate decisions during wound healing, fine-tuning cellular manifestations, such as proliferation, migration, and differentiation [67,68]. Similar to our findings, other researchers have reported that OCT4 is not solely confined to embryonic development but also plays a crucial role in tissue repair and regeneration since it can influence wound healing, including cell proliferation, migration, and differentiation. Thereafter, OCT4-stimulating strategies might open promising avenues for therapeutic interventions to enhance wound healing in diverse clinical settings [69,70]. 

Various parts of *C. aspersa* and other mollusks have been documented in a variety of studies to possess potent antioxidant abilities [30,33]. A recent study showed an ability of snail mucus to reduce intracellular ROS levels and therefore protect cells from oxidative stress [33]. Similar results regarding the antioxidant abilities of eggs derived from a different kind of mollusk (*C. aspersa* maxima) were obtained by Gorka and colleagues, using cell-free methods, such as ABTS+ scavenging activity [22]. Furthermore, another study revealed the antioxidant abilities of egg extracts derived from mollusks on colon cancer cells. Additionally, they have documented that eggs derived from *C. aspersa* exhibited an increase in GSH, which was lower than the one from the extract from the eggs derived from *C. aspersa* maxima [36].

In several studies, cellular senescence has been associated with an increase in intracellular ROS levels [13,71,72]. Even more, we have previously demonstrated that as WJ-MSCs age, the augmentation of oxidative stress results in an increase in DNA double-strand breaks [51]. The cells used in the current study have undergone a certain number of passages; therefore, we suspected that they would have accumulated a certain amount of DNA damage induced by a variety of stressors during cell culture, including oxidative stress [12]. CAEE demonstrated a strong protective function against oxidative DNA damage formation. Similar to this, Espada and colleagues recently showed that *C. aspersa* egg extract can mitigate UVB-induced DNA damage in a HaCaT cell line [34], strengthening our conclusion of its protective effect against DNA damage. Espada and colleagues have also provided evidence of the anti-senescent properties of *C. aspersa* eggs. Specifically, they demonstrated that treatment of senescent human diploid fibroblast (SHDFs) with *C. aspersa* eggs can reduce several senescence-associated markers, such as the number of SA-β-gal positive cells and the levels of senescence markers p16 and p53 [34]. Here, we demonstrated that CAEE could even protect against premature senescence that is exogenously induced by acute oxidative insult. Stress-induced premature senescence (SIPS) is a phenomenon that is characterised by accelerated ageing and impaired regenerative capacity, which by definition hinders the therapeutic potential of MSCs. 

Autophagy is a cellular process mediated by lysosomes that leads to the degradation of cell content under starvation or other stressful conditions, such as senescence [73,74]. However, autophagy’s relationship with cellular senescence remains ambiguous as its main role in removing damaged organelles promotes cellular homeostasis and is therefore protective against senescence, whereas its involvement in the biogenesis of certain senescence-associated secretory phenotype (SASP) factors promotes senescence [75,76]. On the other hand, oxidative stress has a strong relationship with autophagy as it constitutes one of its main inducers [71]. Autophagy is also closely related to DNA damage. As reviewed by Filomeni and colleagues, DNA damage is sensed primarily by the proteins PARP1 and ATM. Subsequently, their signaling pathway induces AMPK activation, a molecule that acts as a positive regulator of autophagy, allowing cells to remove any damaged compounds [74]. Given the fact that during oxidative SIPS cells accumulate DNA damage, it is reasonable to anticipate an increase in the autophagic flux. Therefore, the protective ability of CAEE that we have described here against DNA damage and subsequently from oxidative SIPS could be explained by its ability to induce autophagy activation. 

We have previously revealed that in response to oxidative stress Caveolin-1 is upregulated, phosphorylated, and translocated into the nucleus, where is involved in oxidative DNA damage repair [51]. In line with the above, we revealed here that Caveolin-1 levels after CAEE treatment are diminished, possibly because of less DNA damage formation and therefore lack of the necessity for Caveolin-1-mediated DNA damage repair. Even more, in the same previous report of our group [51], we demonstrated that downregulation of Caveolin-1 with siRNA results in autophagy induction. As a matter of fact, extensive research detailing the intricate relationship between Caveolin-1 and autophagy is available. Caveolin-1 appears to be both directly [53,55,77] and indirectly [54] connected to autophagy. Here, we reveal the ability of snail egg extract to control the intracellular levels of Caveolin-1 that could serve as the means of regulating autophagy activation. 

In conclusion, our results demonstrate that *Cryptomphalus aspersa* egg extract fortifies the stemness of WJ-MSCs, markedly ameliorates their wound-healing abilities, increases their antioxidant potential, and therefore protects them against oxidative DNA damage. More importantly, we showcased that CAEE treatment prevents stress-induced premature senescence possibly through regulation of Caveolin-1/autophagy interplay. The latter could not only provide a tool necessary for clarifying the pleiotropic function of Caveolin-1 in senescence but could also contribute to overcoming the obstacles related to MSCs during in vitro propagation and therefore establishing them as a crucial means of regenerative medicine applications. 

## 4. Material and Methods

### 4.1. Primary Cultures of Mesenchymal Stem Cells (MSCs)

MSCs were extracted from the Wharton jelly of the umbilical cord (WJ-MSCs) of neonates after birth upon parental authorization (three distinct entities, n = 3), as previously described [78]. All the isolated WJ-MSCs were propagated in standard culture conditions at 37 °C and 5% CO_2,_ in Dulbecco’s modified Eagle’s medium DMEM containing high glucose as well as stable glutamine and sodium pyruvate (BioWest, Miami, FL, USA) with the inclusion of 10% fetal bovine serum (FBS; Thermo Fisher Scientific, Waltham, MA, USA) and 1% penicillin-streptomycin (Thermo Fisher Scientific, Waltham, MA, USA). Culture medium was changed, and cells were passed when they reached 80% confluence, and fresh complete medium was used. Middle-passage cells (20 < *p* < 25) were used in all experiments of this study.

### 4.2. Animal Farming and Preparation of the Egg Extracts

Fresh *Cryptomphalus aspersa* snail eggs were obtained from a greenhouse-type snail farm located in Central Greece (Prefecture of Thessaly) [79]. The protocol to create the egg extract (CAEE) followed here was similar to that described in Matusiewicz and colleagues [36]. In detail, shortly after harvesting, the eggs were washed, homogenised, and frozen at −80 °C, for 2 days. The eggs then underwent subsequent lyophilisation for 3 days. Crushing of the lyophilised eggs using a laboratory pestle ensued until they formed a powder. CAEE preparation was conducted once, and all lyophilised aliquots were kept in polypropylene tubes at ambient room condition until reconstitution.

For the reconstitution of CAEE, in order for it to be used in the experiments, we followed a previously described methodology [40]. In brief, the fine powder of lyophilised eggs was homogenised in deionised water to generate an initial stock solution (100 mg/mL). Vigorous vortexing ensued, followed by extraction (30 min, 4 °C), centrifugation (1600× *g*, 10 min), and collection of the supernatant, which served as the stock solution extract. The extract was then passed through a polyvinylidene fluoride (PVDF) syringe filter (pore size 0.22 μm; EuroClone, Pero, Italy) and stored at −80 °C until used for experimentation purposes. Every aliquot was used once for each experiment and then discarded.

### 4.3. Stress-Induced Premature Senescence (SIPS)

Firstly, cells were grown in culture flasks and upon reaching 60–70% confluence were treated or not with CAEE 0.1 mg/mL for 24 h. Afterwards, oxidative stress-induced premature senescence (SIPS) was achieved using a previously described methodology [51]. Cells at approximately 80% confluency were subjected to 400 μM of H_2_O_2_ for 2 h in DMEM without FBS. Subsequently, cells were washed twice with phosphate buffer saline (PBS) and then incubated in normal complete medium. Cells were divided in a 1:2 ratio after 1 to 3 days (passage 1—after treatment—SIPS p1), and after 3 to 7 more days (at 80% confluency), an additional split was made (passage 2—following treatment SIPS p2). After that, cells were seeded in 6-well plates for protein extraction or SA-β-gal staining. 

### 4.4. Protein Extraction and Western Blot Analyses

Cells were collected using trypsin and whole-cell lysates were created by mixing with RIPA lysis buffer [10 mM Tris (pH 7.5), 150 mM NaCl, 1% Triton X-100, 1% sodium deoxycholated, 0.1% SDS, 1 mM EDTA] supplemented with protease and phosphatase inhibitors (Thermo Fisher Scientific, Waltham, MA, USA). Subsequently, lysates were kept on ice for 30 min and thoroughly vortexed every 5 min. Cell lysates were then centrifuged for 15 min at 4 °C at 12,000 rpm, and the respective supernatants were obtained. Total protein quantification was performed using the Pierce^TM^ BCA Protein Assay kit (Thermo Fisher Scientific, Waltham, MA, USA). Twenty (20) μg of total protein were separated into 10% or 12% sodium dodecyl sulfate-polyacrylamide gel electrophoresis gels (SDS-PAGE) and then transferred to polyvinylidene fluoride (PVDF) membranes (Thermo Fisher Scientific, Waltham, MA, USA). For membrane blocking, 5% *w*/*v* non-fat dry milk in TBS/0.1% Tween^20^ or 5% BSA in PBS/0.1% Tween^20^ was freshly prepared and used for incubation of the membranes for 1 h at 4 °C. After blocking, the membranes were incubated overnight at 4 °C with specific primary antibodies against p21 (1:1000, Cell Signaling Technology, Danvers, MA, USA, Rabbit #2947), p16 (Cell Signaling Technology, MA, USA, Rabbit, D3W8G), LC3A/B (1:1000 dilution, Cell Signaling Technology, MA, USA, Rabbit #4108), SQSTM1/p62 (1:1000 dilution, Cell Signaling Technology, MA, USA, Mouse #88588), Beclin-1 (1:1000 dilution, Cell Signaling Technology, MA, USA, Rabbit #3495), PI3K-CIII (1:1000 dilution, Cell Signaling Technology, MA, USA Rabbit #4263), ATG5 (1:1000 dilution, Cell Signaling Technology, MA, USA, Rabbit #12994), ULK-1 (1:1000 dilution, Cell Signaling Technology, MA, USA, Rabbit #8054), Caveolin-1 (1:1000 dilution, Cell Signaling Technology, MA, USA, Caveolin-1 Rabbit #3251), NRF2 (1:1000 dilution, Cell Signaling Technology, MA, USA, Rabbit #D1Z9C), NQO1 (1:1000 dilution, Cell Signaling Technology, MA, USA, Mouse #A180), OCT4 (1:1000 dilution, Cell Signaling Technology, MA, USA, Rabbit #C30A3), and β-actin (1:1000 dilution, Cell Signaling Technology, MA, USA, Mouse #8H10D10) or α-tubulin (1:1000 dilution, Cell Signaling Technology, MA, USA, Rabbit #11H10), which served as the loading controls. On the following day, the membranes were washed three times for 10 min with TBS/0.1% Tween^20^ and were gently agitated before they were incubated for 1 h with the proper horseradish peroxidase (HRP)-conjugated secondary antibodies at room temperature (RT): anti-rabbit (1:10,000 dilution, Boster, CA, USA #BA1054-1) and anti-mouse (1:10,000 dilution, #BA1050-1, Boster, CA, USA). All protein bands were visualised using ECL substrates (Thermo Fisher Scientific, Waltham, MA, USA) and detected by Uvitec Cambridge Chemiluminescence Imaging System. For the analysis of protein expression, Image J software (1.47r, Wayne Rasband National Institutes of Health, USA) was used. Experimental control group protein levels’ value was arbitrarily set to 1. 

### 4.5. Senescence-Associated β-Galactosidase (SA-β-Gal) Staining

Approximately 2 × 10^5^ cells were seeded in 6-well plates as described in paragraph 4.3 and subsequently rinsed/washed 3 times with cold PBS prior to fixation with a mixture of 2% formaldehyde/0.2% glutaraldehyde at room temperature (24–26 °C) for 5 min. Following fixation, 3 additional cold PBS washes were performed before the staining step. For staining, the cells were incubated at 37 °C without the presence of CO_2_ overnight with a freshly made SA-β-gal staining solution (40 mM citric acid/sodium phosphate buffer pH 6.0, 5 mM potassium ferrocyanide, 5 mM potassium ferricyanide, 150 mM NaCl, 2 mM MgCl_2_, 1 mg/mL 5-bromo-4-chloro-3-indolylβ-D-galactoside). For data acquisition, two independent researchers enumerated at least 300 cells, and the amount of SA-β-gal positive cells was represented as the percentage of the total cells observed. The SA-β-gal assay was repeated three times with WJ-MSCs from three different donors (n = 3).

### 4.6. Immunofluorescence (IF)

Immunofluorescence tests were carried out as previously described [51]. Specifically, 10^5^ cells were cultured in 6-well plates with coverslips and upon reaching 80% confluency were treated or not with CAEE 0.1 mg/mL for 24 h. The cells were afterwards fixed with ice-cold absolute methanol at −20 °C for 10 min. Coverslips were blocked in PBS containing 0.2% Tween^20^ and 1% BSA for 10 min, then incubated for 1 h at room temperature, with specific primary antibody against 53BP1 (1:500, clone BP13, mouse monoclonal, Millipore, MA, USA) for double-strand breaks (DSB) detection and the appropriate secondary antibody (1:500 dilution, Alexa Fluor 488, Molecular Probes). For monitoring autophagy activity by immunofluorescence, we used antibodies against LC3 and p62, as previously described [80]. In detail, cells on coverslips treated as above were incubated with specific primary antibody against LC3A/B (1:200 dilution, Cell Signaling Technology, MA, USA, Rabbit #4108) as well as with specific primary antibody against p62 (1:200 dilution, Cell Signaling Technology, MA, USA, Mouse #88588) and the appropriate secondary antibodies (1:500 dilution, Alexa Fluor 488, Molecular Probes for p62 and 1:500 dilution, Alexa Fluor 594, Molecular Probes for LC3 A/B). After that, 4,6-diamidino-2-phenylindole (DAPI)-containing Vectashield mounting media (Vector Laboratories, Burlingame, CA, USA) was used to visualise the nuclei. 

Images were acquired using a ZEISS Axio Imager.Z2 fluorescence microscope at 40× or 100× magnification (depended on the experiment) and analysed with ImageJ software. For each treatment, two independent observers, blinded to the sample’s origin, examined at least 5 randomly chosen fields, measuring at least 100 cells for each condition. The means of their counts were used to determine statistical significance. The experiments were repeated at least three times.

For the quantification of autophagy induction, the percentage of cells positive for distinct LC3 puncta was calculated by dividing the number of cells which exhibited large LC3 puncta by the total number of cells in each field. At least five different fields were observed for each condition by two independent researchers, and at least 100 different cells were counted. The means of their counts were used to determine statistical significance. The experiments were repeated at least 3 times. Images were acquired using a ZEISS Axio Imager.Z2 fluorescence microscope at 100× magnification and analysed with ImageJ software.

### 4.7. Cell-Migration Assay

For the cell-migration assay (or wound-healing assay), WJ-MSCs were seeded in 6-well plates and cultured under optimal conditions until they reached 90% confluence. As previously described [80], without removing the culture medium, scratches (“wounds”) were made manually in a confluent monolayer of cells using a 1000 μL pipette tip. The cells were subsequently washed with phosphate buffer saline (PBS) in order to remove any floating cells, and different CAEE concentrations (25 mg/mL, 2.5 mg/mL, 1 mg/mL, 0.1 mg/mL) were added. Experimental control group cells were treated with an additional complete medium. The cells were observed under an inverted microscope to assess their healing state, and photographs were taken right after wound infliction (t = 0) and 24 h after (t = 24 h). The experiments were replicated at least three times.

### 4.8. Determination of GSH Levels

WJ-MSCs were trypsinised, collected, and centrifuged at 15,000× *g* for 5 min at room temperature. Subsequently, the cell pellet was resuspended in PBS, followed by mechanical lysis of the cells through rigorous vortexing. Bradford assay was employed to determine the total protein concentration.

GSH levels were determined according to Kolonas and colleagues [81]. Briefly, 50 μL of cell lysate was mixed with 67 mM of potassium phosphate buffer (pH 8) and freshly made DTNB [5,5′-dithiobis-2 nitrobenzoate] dissolved in 1 mM in 1% sodium citrate and vortexed. The samples were then incubated in the dark for 15 min at room temperature, and the optical density was determined at 412 nm. For the calculation of GSH levels, the molar extinction co-efficient of DTNB was used, and GSH concentration was expressed as nmol of GSH per mg of protein. GSH levels are depicted as % of control. Each experiment was repeated at least three times.

### 4.9. TBARS Assay for the Determination of Lipid Peroxidation

TBARS assay used here was described by Kolonas and colleagues [81]. All initial steps of the procedure are the same as in Section 4.8. After cell lysis and protein concentration determination, 30 μg of total protein was used for all samples, whereas PBS was used for the blank. Tris-HCL (200 mM, pH 7.4) and 35% TCA were added in a 1:1 ratio, and the samples were incubated at room temperature for 10 min. Next, a solution of 2 M Na_2_SO_4_ and 55 mM thiobarbituric acid (TBA) was added, followed by an incubation at 95 °C for a total of 45 min. Afterwards, the samples were left to cool on ice for 5 min, and 1 mL of 70% TCA was added. The samples were once again rigorously vortexed and centrifuged at 15,000× *g* for 3 min. Finally, the absorbance of the supernatants was carefully measured at 530 nm. For the calculation of TBARS concentration, the molar extinction co-efficient of malondialdehyde was used, and TBARS was expressed as nmol of per mg of total protein. TBARS levels are depicted as % of control. Each experiment was repeated at least three times.

### 4.10. Statistical Analysis

For data analysis, Graph Pad Prism 8.0.1 software was employed. Student’s *t*-test or one-way ANOVA was used to establish statistical significance. P values less than 0.05 were regarded as statistically significant for all comparisons. The significance on the graphs is indicated with asterisks (* = *p* < 0.05, ** = *p* < 0.01, *** = *p* < 0.001, **** = *p* < 0.0001) for correlation between each treatment and no-treatment (NT) condition or otherwise indicated. The data are presented as mean ± standard error (means ± S.E.).

## Figures and Tables

**Figure 1 ijms-25-03715-f001:**
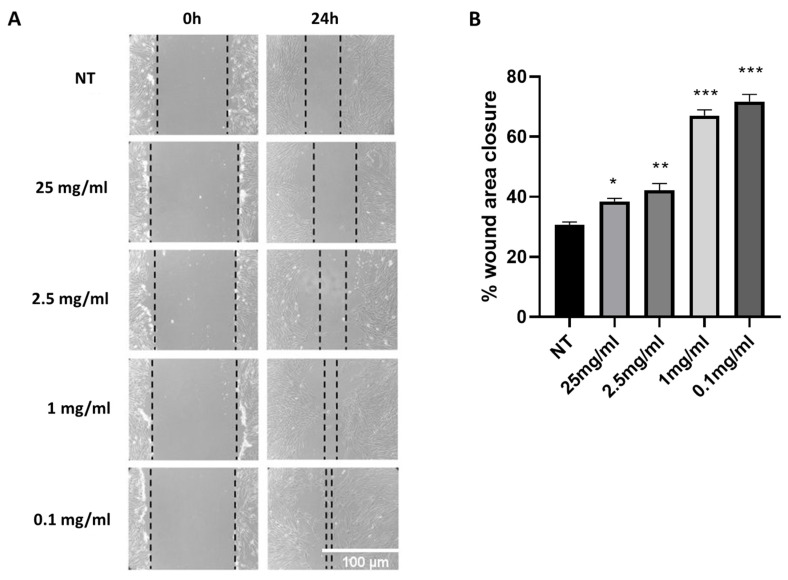
*Cryptomphalus aspersa* egg extract (CAEE) exhibits potent wound-healing ability. (**A**) Representative images of “wounds” treated with various concentrations of the egg extract (CAEE) or “wounds” that have not been treated with the extract (NT), immediately after “wound” infliction (0 h) and 24 h post “wound” infliction (24 h). (**B**) Graph demonstrating “would healing” expressed as the percentage of original “wound” area closure after 24 h for “wounds” treated with various concentrations of the extract (CAEE) and “wounds” not treated with the extract (NT). Values shown are the means ± S.E. from three different experiments with WJ-MSCs derived from three different donors (n = 3) treated (CAEE) or untreated (NT). Asterisk marks statistical significance (*p* < 0.05) in all panels (* = *p* < 0.05, ** = *p* < 0.01, *** = *p* < 0.001).

**Figure 2 ijms-25-03715-f002:**
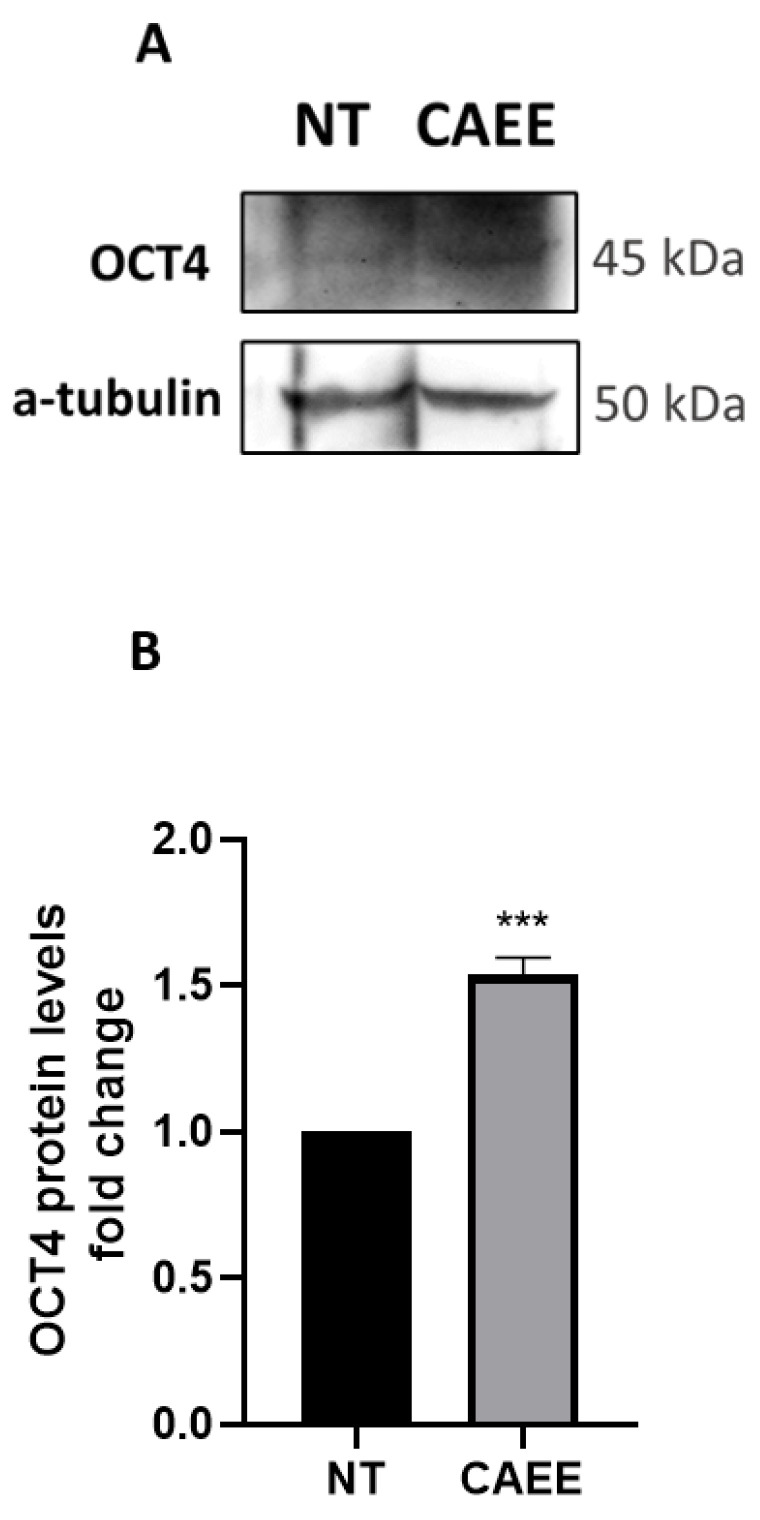
Treatment with CAEE increases OCT4 protein levels. (**A**) Representative immunoblot and (**B**) densitometric analysis of OCT4 protein levels (fold change) in cells that have been treated with CAEE (for 24 h) or left untreated (NT). A-tubulin served as a loading control. NT value was arbitrarily set to 1. The values shown are the means ± S.E. as the experiment was repeated 3 times using cells from three different donors (n = 3). Asterisk marks statistical significance (*p* < 0.05) in all panels (*** = *p* < 0.001).

**Figure 3 ijms-25-03715-f003:**
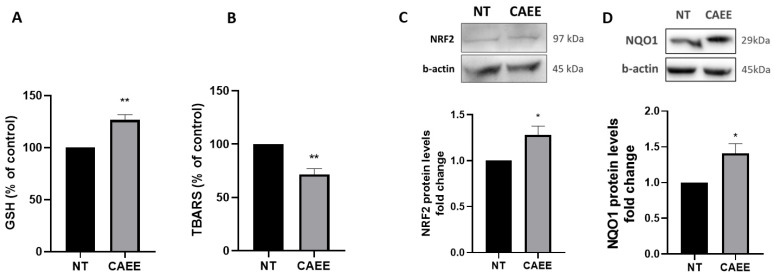
CAEE effect on the antioxidant potential of WJ-MSCs. (**A**) Bar graph demonstrating the change of glutathione (GSH) levels expressed as % of control levels measured in WJ-MSCs treated with CAEE for 24 h or not (NT). NT value was arbitrarily set to 100. (**B**) Bar graph demonstrating the change in TBARS levels expressed as % of control levels measured in WJ-MSCs treated with CAEE for 24 h or not (NT). (**C**,**D**) Representative immunoblots and densitometric analysis of NRF2 (**C**) and NQO1 (**D**) protein levels (fold change) in WJ-MSCs treated with CAEE for 24 h compared to untreated WJ-MSCs (NT). NT value was arbitrarily set to 1. The values shown are the means ± S.E. as the experiments were repeated 3 times using cells from three different donors (n = 3). Asterisk marks statistical significance (*p* < 0.05) in all panels (* = *p* < 0.05, ** = *p* < 0.01).

**Figure 4 ijms-25-03715-f004:**
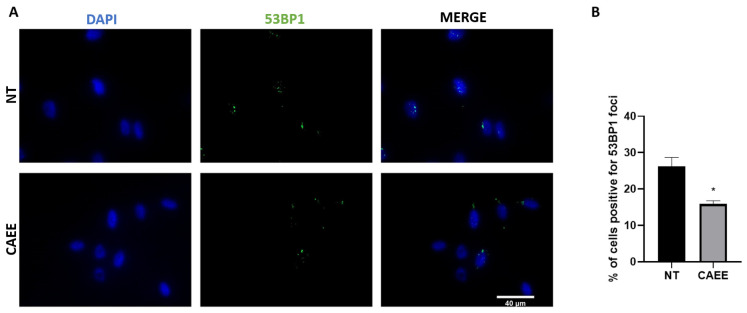
Treatment with CAEE significantly reduces DNA damage. (**A**) Representative images illustrating DNA damage assessed by staining with anti-53BP1 antibody (green) in WJ-MSCs treated with CAEE or not (NT) where nuclei were visualised using DAPI (blue). Images were acquired using fluorescence microscope (40× objective lens). (**B**) Bar chart depicts the percentage of cells with DNA damage (cells with 4 or more 53BP1 foci) per condition (treated with CAEE or NT). The experiment was repeated three times with cells from 3 different donors, and each time at least 5 different fields were examined for each condition, measuring up to at least 100 different cells. The values depicted are the means ± S.E. Asterisk marks statistical significance (*p* < 0.05) in all panels (* = *p* < 0.05).

**Figure 5 ijms-25-03715-f005:**
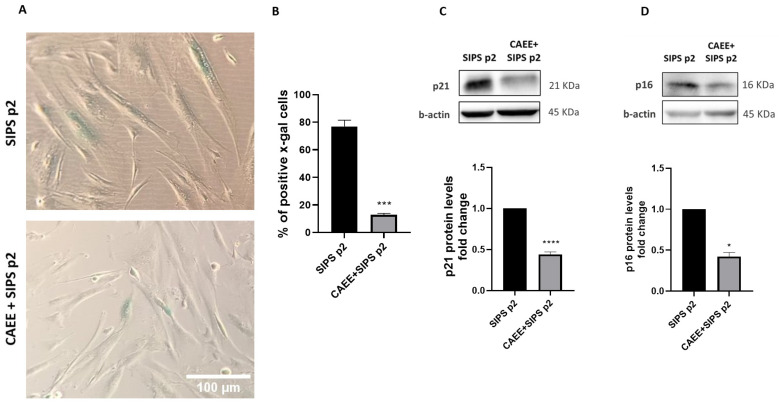
CAEE treatment protects against oxidative stress-induced premature senescence (SIPS). (**A**) Representative photos of SA-β-gal staining of WJ-MSCs treated with 400 μΜ H_2_O_2_ for 2 h to induce premature senescence at passage 2 (SIPS p2) and WJ-MSCs treated with CAEE (for 24 h) prior to senescence induction (CAEE+SIPS p2). (**B**) Bar graph demonstrating the mean percentages ± S.E. of SA-β-gal positive cells (blue). Two independent researchers enumerated at least 300 cells, and the amount of SA-β-gal positive cells was represented as the percentage of the total cells observed. The experiment was repeated three times with cells from 3 different donors (n = 3) and values shown are the means ± S.E. (**C**,**D**) Representative immunoblots and respective densitometric analysis of p21 and p16 protein levels (fold change) in WJ-MSCs treated with 400 μΜ H_2_O_2_ for 2 h to induce premature senescence at passage 2 (SIPS p2) and WJ-MSCs treated with CAEE (for 24 h) prior to senescence induction (CAEE+SIPS p2). The values shown are the means ± S.E. as the experiments were repeated three times with cells from 3 different donors (n = 3). SIPS p2 value was arbitrarily set to 1. Asterisk marks statistical significance (*p* < 0.05) in all panels (* = *p* < 0.05, *** = *p* < 0.001, **** = *p* < 0.0001).

**Figure 6 ijms-25-03715-f006:**
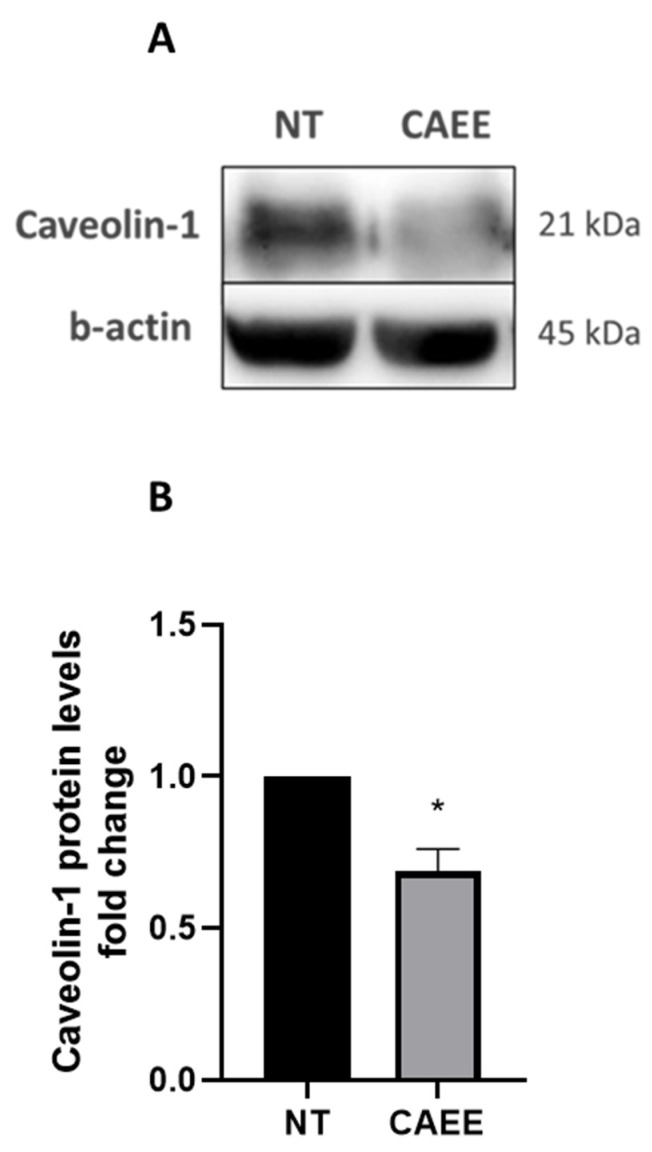
CAEE treatment decreases Caveolin-1 protein levels in WJ-MSCs. Representative immunoblots (**A**) and densitometric analysis (**B**) of Caveolin-1 protein levels (fold change) in WJ-MSCs treated with CAEE for 24 h compared to untreated WJ-MSCs (NT). The values depicted are the means ± S.E. from three different experiments with cells from three different donors (n = 3). NT value was arbitrarily set to 1. Asterisk marks statistical significance (*p* < 0.05) in all panels (* = *p* < 0.05).

**Figure 7 ijms-25-03715-f007:**
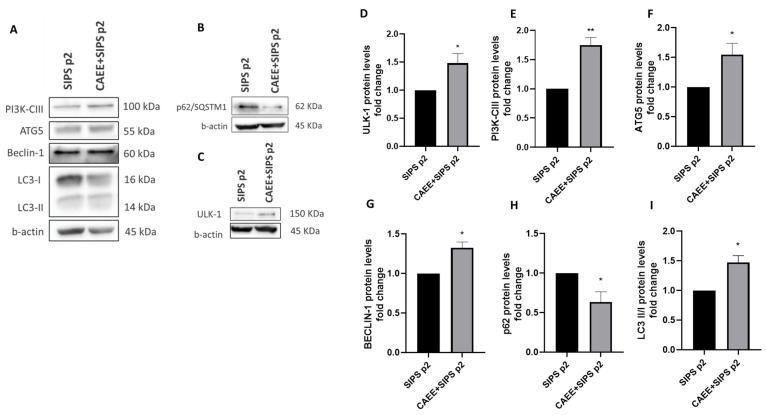
CAEE treatment stimulates autophagy to counteract SIPS induction. (**A**) Representative immunoblots of autophagy-related proteins in WJ-MSCs treated with 400 μΜ H_2_O_2_ for 2 h to induce premature senescence at passage 2 (SIPS p2) and WJ-MSCs treated with CAEE (for 24 h) prior to senescence induction (CAEE+SIPS p2). (**B**,**C**) Representative immunoblots of p62 and ULK-1 proteins in WJ-MSCs treated with 400 μΜ H_2_O_2_ for 2 h to induce premature senescence at passage 2 (SIPS p2) and WJ-MSCs treated with CAEE (for 24 h) prior to senescence induction (CAEE+SIPS p2). Densitometric analysis of ULK-1 (**D**), PI3K-CIII (**E**), ATG5 (**F**), BECLIN-1 (**G**), p62 (**H**), and LC3II/I (**I**) protein levels (fold change) in WJ-MSCs SIPS p2 and WJ-MSCs CAEE+SIPS p2. SIPS p2 value was arbitrarily set to 1. The values depicted are the means ± S.E. from three different experiments with cells from three different donors (n = 3). Asterisk marks statistical significance (*p* < 0.05) in all panels (* = *p* < 0.05, ** = *p* < 0.01).

**Figure 8 ijms-25-03715-f008:**
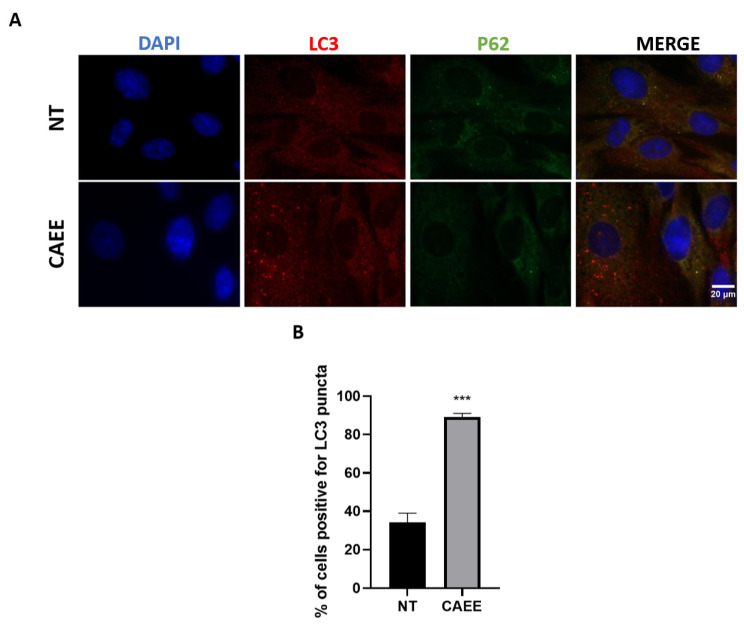
Treatment with CAEE results in autophagosome formation and p62 degradation. (**A**) Representative images of WJ-MSCs treated (CAEE) or untreated (NT) with the egg extract and stained with LC3 (red) and p62 (green). Images were acquired using fluorescence microscope (100× objective lens). (**B**) Bar chart illustrating the percentage of cells positive for LC3 puncta. At least five different fields were observed for each condition, and at least 100 different cells were counted. The values shown are the means of their count ± S.E. from three different experiments with cells from three different donors (n = 3). Asterisk marks statistical significance (*p* < 0.05) in all panels (*** = *p* < 0.001).

## Data Availability

Data are contained within the article and Appendix A. Data are available upon request to the corresponding author.

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
