# Peer review of "Cryptomphalus aspersa Egg Extract Protects against Human Stem Cell Stress-Induced Premature Senescence"

_ijms, 2024, doi:10.3390/ijms25073715_

Round 1

Reviewer 1 Report

Comments and Suggestions for Authors

In the article entitled “Cryptomphalus aspersa Egg Extract as a novel source of bioactive compounds with potent effects against Human Stem Cell Stress Induced Premature Senescence” by Zozo Outskouni and colleagues, the authors show that Cryptomphalus aspersa Egg Extract (CAEE) increases OCT4, GSH, NRF2, NQO1 levels of WJ-MSC, decreases caveolin-1, protects against oxidative stress induced premature senescence. The article is a bit chaotic, but has some interesting points. Perhaps a few of my comments and questions will allow the authors to improve the manuscript.

The CAEE exhibits antioxidant properties by increasing the level of antioxidant molecules. Did the authors measure specific oxidative damage (lipids peroxidation or anything else) or ROS levels?

The authors showed an increase in autophagy proteins. Are there ways to demonstrate changes in autophagy at a functional level?

Figure 1. Why does wound closure decrease with increasing concentrations of CAEE? The authors claim that CAEE enhances wound healing. This is a fundamental point that requires a detailed explanation. Could it be that using even lower concentrations would result in a more pronounced effect?

Figure 2. OCT4 or Oct4?

Figure 4. In this experiment, cells were not treated with H2O2. Why do the authors assume that CAEE protects cells from oxidative stress? In this case, I would suggest that CAEE promotes DNA damage repair.

Line 380-382. In this work the authors did not reveal the ability of snail egg extract to induce autophagy by regulating the intracellular basal levels of Caveolin-1. There is not enough evidence to support the claim. But the authors may assume this.

Number of experiments (n) is unknown. Must be improved.

Minor revision 

Reviewer 2 Report

Comments and Suggestions for Authors

In the manuscript by Outskouni et al, the authors have investigated the effect of  Cryptomphalus aspersa Egg Extract (CAFE) on the MSCs and have shown that it leads to increased stemness; enhances anti-oxidant properties, develops protection against oxidative stress induced premature senescence (SIPS) via its ability to induce autophagy. Overall the topic is interesting and has industrial/commercial potential. However, the use of WT-MSCs and not any disease model is a bit confusing and overall the results presented here are of inferior quality. The wound healing results are confusing and signal to noise ratio in most of their results is pretty low. The supplementary original images are not properly labeled and some of the lanes label are missing.

In the current form, this manuscript is of very poor quality. Here are my major concern and needs a repetition of many of their experiments before I could even consider this for publication.

1.       Figure 1: The results are a bit confusing. Why decreasing concentration has more effect on wound healing? What happens if they went further lower i.e. below 0.1 mg/mL? Did they perform titration experiment to show if the higher concentrations are toxic to cells and below 0.1mg/ml? They should cite what were the previous concentrations used by others.

2.       Figure 2: The Oct4 western is ugly. Their full western blot pictures in supplementary are also not helpful; why the middle lane is not labelled? Why did they use Tubulin which is right next to the Oct4 band; why not some other housekeeping protein of diff. molecular weight so membrane cutting could be more reliable?

3.       Figure 3: Just like Fig 2, their NRF2 western is also not of good quality; so it becomes hard to interpret results. As the signal to noise ratio is very low.

4.       Figure 4: How many times was this experiment repeated? How many cells and replicates were counted? All the statistics should be clearly mentioned and stated. Those images of multiple fields should be shown in supplementary including their cell count pipeline/method. Scale bars are missing.

5.       Figure 5: The magnification in this figure looks totally different.  While cells in top panel appear to be zoomed in and lower panel zoomed out. Scale bar is missing.  Their western blots are of inferior quality. The p16 western is cut in the middle of band; why?  It needs to be repeated.

6.       Figure 6: Another very bad western. They really need to standardize their western blot protocol to get cleaner bands.

7.       Figure 7: again the western results for ULK-1 and p62 definitely don’t qualify for publication. How could one use them to draw scientific conclusion? These needs to be fixed before interpreting the results.

8.       Studying these effects in WT cells is not satisfactory. Even if I ignore their inferior quality western blot and minimal fluorescence microscopy; I fail to wrap my head around some of their experimental design and conclusions. Why not study a disease model cell line?

Comments on the Quality of English Language

Minor editing needed

Reviewer 3 Report

Comments and Suggestions for Authors

The study by Outskouni et al. reports on the beneficial effects of Cryptomphalus aspersa eggs extracts on stem cell migration and senescence. The outcomes of the study are potentially relevant to many researchers in the of stem cell research. However, the manuscript is not well written. The quality of the manuscript and the presented results need substantial improvements.

 Major:

Lines 126-127: The authors performed cell migration assays which were referred to as wound healing assays. This may be misleading. Wound healing assays are performed in animals or composite tissue models where the inflicted wounds affect several layers of tissue and cells. Therefore, the authors may refer to “cell migration assay” and “manually created scratch in a confluent monolayer of cells”.

Line 129: The quality of CAEE should be clarified in detail. Did the authors use a single CAEE preparation for all experiments in the entire study? Did they examine the quality and the quantity of proteins in CAEE fractions? This information is highly relevant to the conclusions.

Figures 2, 3, 5, and 7: All presented immunoblot images are of low quality. The authors are advised to zoom out the presented blot images to authenticate adequate protein signals.

Figures 1A, 4A, and 5A: Scale bars in cell culture images are missing.

Figure 5A: Were both cell culture images taken under identical settings?  The upper panel appears higher magnified. Does % of positive cells refer to a specific size of surfaces? Please explain.

Minor:

The manuscript text needs substantial improvements. Only few examples from the Abstract and the Results sections are listed below. The authors are kindly advised to revise other sections as well.

Lines 1-4: The title is too long and speculative.

Line 18: “Process” is more adequate.

Lines 23-26: Please revise.

Line 27 and 30: Even more is repetitive.  

Lines 41-45: Repetitive, please revise.

Lines 46-49: The provided information is not clear. “Moral dilemmas” is not appropriate.   

Lines 52-55: The provided information is not clear.

Lines 73-75: The provided information is not clear.

Comments on the Quality of English Language

 English very difficult to understand and extensive editing is required.

Round 2

Reviewer 2 Report

Comments and Suggestions for Authors

In the revision of manuscript, the authors have tried to do some new western and edited the manuscript but some of these responses are not satisfactory.  

1.       Regarding my question on CAEE concentrations and would healing; their response is not satisfactory. From the current data it cannot be ruled out the concentrations lower than 0.1 mg/ml could be more optimal unless they have tested it.

2.       The western blot quality of new westerns is still low quality. I do not observe much improvements is some of western.

3.       And if the authors have included new western blots in the manuscript now, how come the quantification bar plots of those western stay the same? They should have quantified these new westerns and replace the old quantifications.

Comments on the Quality of English Language

it's fine.

Reviewer 3 Report

Comments and Suggestions for Authors

I thank the authors for addressing all my concerns in the revised version of the manuscript.

Round 3

Reviewer 2 Report

Comments and Suggestions for Authors

The additional data shown for figure 1 in reviewer's response should be added to supplementary as the results make more sense with these additional data points and their chosen concentration makes sense. The authors have addressed all my concerns now.
